# The Level and Limitations of Physical Activity in Elderly Patients with Diabetes

**DOI:** 10.3390/jcm13216329

**Published:** 2024-10-23

**Authors:** Karolina Biernat, Dominik M. Marciniak, Justyna Mazurek, Natalia Kuciel, Katarzyna Hap, Michał Kisiel, Edyta Sutkowska

**Affiliations:** 1University Rehabilitation Centre, Medical Faculty, Wroclaw Medical University, Borowska 213, 50-556 Wroclaw, Poland; karolina.biernat@umw.edu.pl (K.B.); natalia.kuciel@umw.edu.pl (N.K.); katarzyna.hap@umw.edu.pl (K.H.); 2Department of Dosage Form Technology, Faculty of Pharmacy, Wroclaw Medical University, Borowska 211 A, 50-556 Wroclaw, Poland; dominik.marciniak@umw.edu.pl; 3Medical Faculty, Wroclaw Medical University, 50-367 Wroclaw, Poland; michal.kisiel@student.umw.edu.pl

**Keywords:** diabetes, physical activity, elderly, limitations of physical activity, IPAQ

## Abstract

**Background/Objectives:** Old age and illnesses can limit physical activity (PA). We have assessed the level of PA and its limitations in older people with diabetes (DM). **Methods:** Cross-sectional study, period: January and June 2024, place: Diabetes Center, subjects: patients (N = 207) > 64 years with DM), Surveying using: IPAQ, Accompanying Survey (AS). The significance was assessed by: Student’s *t*-test, Mann–Whitney U test, Pearson’s test. The relationships between the IPAQ and the AS results were analyzed based on a meta-analysis model for variable effects, OR with a 95% CI. ROC curve was used to determine the threshold values for age, BMI. Correlations between selected key variables were evaluated using the PCA. **Results:** The median age: 72 years (65–87), BMI: 28.70 (18.61–49.69). The median PA level: 1837 MET-min/week (0–9.891). The individuals who obtained insufficient (n = 28), sufficient (n = 75), and high levels of PA (n = 53) were 17.95%, 48.08%, and 33.97%, respectively. Diseases were the main factor limiting PA (53.54%) in patient’s opinion, followed by the lack of a companion or motivation. The meta-analysis found no diseases linked to insufficient PA, but those with BMI > 33.3 and over 75 years old more often had insufficient PA. PCA revealed several characteristics that predispose individuals to insufficient PA. **Conclusions:** People over 75 years of age tend to avoid PA more than younger seniors, leading to its insufficient level, similarly like higher BMI. The individual with DM who has an insufficient level of PA is typically a single, woman, over 75, BMI > 33.

## 1. Background

The scientific literature emphasizes the crucial role of physical activity (PA) in maintaining functional ability and overall health among individuals aged 65 and older. PA is broadly defined as any bodily movement generated by skeletal muscles that necessitates energy expenditure [1]. Engaging in regular PA has been shown to decrease the incidence of sarcopenia [2], reduce the frequency of falls [3], lower the risk of cardiovascular disease [4], decrease the number of days spent in hospital [5], and decrease overall mortality rates [6], even among frail older adults.

The clinical advantages of PA are well documented and encompass various aspects, including the attenuation of decline in muscle function and cardiorespiratory fitness, as well as the preservation of functional ability and management of chronic diseases [3,4]. Additionally, from a sociological perspective, PA can contribute to enhanced embodied pleasures in later life, improve overall well-being, facilitate social interactions, alleviate feelings of loneliness, and promote a higher quality of life [7].

Most major guidelines assert that exercise is generally safe for older individuals, and they typically do not need to seek medical advice before increasing their levels of PA [1,8]. However, despite the well-established benefits of PA, older adults tend to fall short of meeting the recommended weekly targets compared to younger age groups [9].

Diabetes mellitus (DM) is a prevalent health condition among the elderly population and has a bidirectional relationship with the level of PA [10]. Over one quarter of individuals aged 65 and above are affected by diabetes, and approximately half of older adults experience prediabetes [11]. The prevalence of these conditions among older adults is anticipated to rise significantly in the coming decades. PA is advocated as a pivotal strategy not only for the prevention but also for managing DM [12]. Elevated levels of PA are correlated with a more favorable metabolic profile and reduced risk of cardiovascular diseases among individuals with DM [13]. Notably, the American Diabetes Association (ADA) recommends engaging in moderate-to-vigorous aerobic exercise for at least 150 min per week [14]. Some researchers propose incorporating alternating bouts of low- and high-intensity exercise to enhance glycemic control and improve endothelial function [14]. Intriguingly, studies comparing various models of PA, encompassing different intensities and durations, indicate that all types, albeit in varying ways, effectively contribute to glycemic and blood pressure control, insulin sensitivity, body composition, muscular strength, and aerobic capacity enhancement [15].

Despite the favorable association between PA and DM management in the elderly population, it remains uncertain what proportion of individuals aged at least 65 and who suffer from diabetes have sufficient PA for their health. In recent years, many studies have addressed the limitations that can restrict PA among the elderly [16], but without details regarding the elderly with diabetes. Hence, as the elderly population with diabetes is growing as well as aging, and PA is still the main therapeutic recommendation, thus it is crucial to assess the local population’s level of PA to comprehend the factors influencing PA engagement among older adults. It is the basis for facilitating the development and implementation of effective strategies to improve PA participation and adherence.

The purpose of the study was to assess the level of PA among the elderly who suffer from diabetes and to determine the main problems that this group encounters when wanting to increase their PA.

## 2. Methodology

### 2.1. The Surveys Used

The IPAQ-International Physical Activity Questionnaire/Polish /Short Form (IPAQ/PL/SF) [17] and the Accompanying Survey (AS) (Appendix A), as a self-rated tool, were used to determine the PA level in a quasi-objective and subjective way, respectively. The demographic and epidemiological data, and questions regarding the patient’s limitations in undertaking regular PA, were obtained from the AS. The following demographic and epidemiological data from the AS survey were analyzed: biological sex, age, education level (primary, secondary, higher), recent occupation (or disability pension, retirement pension), place of residence (name of the town), marital status (married, single—divorced, widowed, unmarried), number of people living with the respondent in the household, weight, height, certified (or any) degree of disability (yes/no), long-term use of medical supplies (yes—specify which/no), HbA1c value, chronic diseases the patient suffers from (multiple choice, please see Appendix A), and whether the patient takes medication for these diseases on a chronic basis (yes/no/yes, but only for some of the above diseases). Patients were then asked to subjectively assess their level of daily physical activity (low-insufficient, sufficient, high) and to indicate whether they would like to be more active (yes/no). If the answer was “yes”, the patient was asked where they would prefer to engage in physical activity: at home or outside the home (they could select both options if the location did not matter). The following questions concerned daily obstacles that, in the patient’s opinion, limit his/her PA (multiple choice allowed). One month of community consultation at the Diabetic Center preceded the selection of the proposed list of limitations to create this last part of the AS (please see the group of questions Q2 in Appendix A, the first column in the table). In the final part, patients were asked to indicate whether, in their opinion, the following diseases/conditions could be a contra-indication to being physically active (please see the group of questions Q3 in Appendix A, the first column in the table).

### 2.2. How and from Whom Were the Data Collected?

Between January and June 2024 (six months), each patient who visited the main investigator, a diabetologist, at the Diabetic Center and met the inclusion criteria, was encouraged to participate in the study. Inclusion criteria: >64 years of age, any sex and race, and any duration and type of diabetes. Exclusion criteria: a recent history of an acute medical condition that may interfere with the reliability of the IPAQ result because the questionnaire assesses the activity from the last seven days (e.g., pneumonia, myocardial infarction (MI), limb injury); an inability to complete the IPAQ or AS questionnaire for any reason as assessed by the interviewer. The final exclusion criterion was defined as a deep mental or physical disability resulting in the misunderstanding of questions or complete inability to perform PA, e.g., in the course of a severe stroke or the active phase of a diabetic foot.

In accordance with the Bioethics Committee decision (approval No.: KB 8/2023), the patient’s verbal consent was sufficient to take the set of questionnaires and answer the questions after reviewing the Study Information document. The patients could fill out the questionnaires in the center (it took about 30 min), but they could also take them home and return them within a week to one of two pointed places. This decision was made because some patients did not have glasses or time to complete the questionnaires at the center. The information was collected anonymously, so each patient received a consecutive number, the same for each questionnaire to par the documents for statistical purposes.

The patient entered height and weight values into the questionnaire themselves, while the researcher calculated body mass index (BMI) after collecting the questionnaire from the patient. The researcher also calculated HbA1c values from percent into mmol/mL as the local laboratory primarily provides percent units, and these were given by the patients. If the patient improperly entered the name of the disease because he/she could not choose it from the proposed ones, the researcher suited it to the proposed one from the questionnaire. For example, in the questionnaire section devoted to disease, “hip degeneration” was written by the patient instead of marking the box with the proposed “diseases of the musculoskeletal system”. In such a situation, the researcher marked this description as “diseases of the musculoskeletal system” in the Excel document prepared for the statistician to provide uniformly defined data from the whole surveyed group.

It was emphasized to the patient that the last part of the AS survey (opinions on the possibility of undertaking PA in chronic disease) is not devoted only to the patient’s disease but generally to his/her idea of what might be an obstacle, if any.

If unclear, the patient could ask the doctor for clarification about some survey questions, except for the part dedicated to the limitation of physical activity, as this part was raised from the previously mentioned population’s consultation.

To cope with the heterogeneity in AS questions, we dichotomized the variable into the categories: parts A, B, and C (please see Appendix A).

### 2.3. Statistics

In this study, variables on various measurement scales were analyzed. Continuous variables such as age, BMI, HbA1c, and MET were expressed in quotient scales. Other variables were nominal, including nominal, directional, and dichotomous. For the continuous variables, the conformity to the normal distribution was assessed with the Shapiro–Wilk test and the homogeneity of variance with the Levene and Brown–Forsyth tests.

Continuous variables were characterized by basic descriptive statistics, including mean value, standard deviation, and 95% confidence interval for normal distribution variables. Variables without a normal distribution were described by median, minimum, and maximum values. Tables of counts, along with percentages and cumulative percentages of the total, were calculated for nominal variables.

In assessing the statistical significance of differences between the mean values of continuous variables in the compared groups, based on the Shapiro–Wilk test result, the Student’s *t*-test was used for independent samples for variables with a normal distribution and the non-parametric Mann–Whitney U test for variables that did not meet this criterion.

The statistical significance of correlations between variables on nominal scales was assessed using the non-parametric chi2-Pearson test. Correlations between the IPAQ variable subjected to categorization into a dichotomous variable (0 vs. 1 or 2) and the results of the three parts of the AS (marked as Q1, Q2, Q3) were expressed on a forest plot and analyzed using the meta-analysis model for the effects of the variables, using the odds ratio with a 95% confidence interval.

ROC curve analysis along with the Youden index was also used in the study to determine the limits of the subjects’ age and BMI index, allowing for their maximal differentiation based on the IPAQ variable subjected to categorization into a dichotomous variable (as above).

The correlations between the selected most significant variables were initially assessed using generalized principal component analysis (PCA). The constructed PCA model was estimated using the NIPALS iterative algorithm. The convergence criterion was set at the level of 0.00001, with the maximum number of iterations equal to 50. The number of components was determined through the maximum of the predictive ability of Q^2^ using the V-fold cross-check method. The resulting optimal PCA model was reduced to two principal components.

For all statistical analyses performed, a significance level of α = 0.05 was assumed. Statistical analysis was performed using STATISTICA PL^®^ version 13.

## 3. Results

Between 1 January and 30 June 2024, 207 individuals filled out both questionnaires. Nevertheless, not all of the questions from the AS were answered, and not each individual fulfilled the IPAQ. For statistical analysis, all of the AS results were analyzed, but if some of the questions were omitted by the patient, this was taken into consideration. It was illustrated in Appendix A (from Appendix A): parts A, B, C as N and ∑N with % and ∑%, respectively. The authors found this maneuver more reliable for further analysis and conclusions than calculating each dataset for 207 sets of questionnaires.

Regarding the IPAQ, we received 172 surveys, but 16 were excluded because they did not meet the criteria for reliable completion suggested in the guidelines [18].

### 3.1. Data Analysis: Appendix A (Appendix A), Part A—Basic Descriptives

Most demographic and epidemiological data are presented in Appendix A—Appendix A, Part A. Women dominated in the studied group. The median age was 72 years (range: 65–87), weight was 80 kg (range: 42.5–147), and BMI was 28.70 (range: 18.61–49.69). All of the participants suffered from diabetes, but the value of HbA1 was only known for 136 (65.7%) participants. The median value of HbA1c was 6.4%; 46 mmol/mol (range 3.76–12.70% and 18–115 mmol/mol). Insulin was taken by 29 individuals (76 denied having insulin therapy), but only 105 patients answered this question.

Both sexes were homogeneous in terms of basic descriptive parameters.

The most common disease was hypertension, followed by musculoskeletal system diseases, gastrointestinal diseases, kidney and venous diseases in similar percentages, thyroid gland disease, and then atrial fibrillation. The least indicated were mental disorders (Appendix A—Appendix A Part A).

Compared to men, women were more likely to suffer from thyroid gland disease (*p* = 0.005), lung, or bronchial disease diseases (*p* = 0.009), musculoskeletal system diseases (*p* = 0.016), and chronic venous insufficiency (*p* = 0.03). However, males more often declared past myocardial infarction (*p* = 0.016) and several diseases grouped as one: kidney, prostate gland, urinary bladder disease (*p* = 0.0003). However, when considering all reported comorbidities, there were no significant differences between women and men.

### 3.2. Data Analysis: Appendix A (Appendix A), Part B—Self-Reported Level of PA and Patient’s Intention

Slightly over 2.5% of the patients scored their activity level as high. Based on information from the AS, similar percentages of responders rated their PA as sufficient (about 50%) or insufficient (about 45%). This self-assessment was completed by 191 survey participants.

Almost 90% of patients who answered the question about whether they wanted to increase their level of PA (N = 167) confirmed that they wanted to be more physically active. More than half of the patients declared that they preferred to be active outside of the home or that the choice of place was not important to them.

The mean BMI for people who preferred to be active outside the home was 28.67 compared to those who chose in-home activity: 30.81 (*p* = 0.023, Student’s *t*-test). Receiver operating characteristic (ROC) analysis determined that a BMI value of 28 was the differentiating point, taking into account the preferred place for PA; people with a higher BMI were more likely to be active inside the home.

### 3.3. Data Analysis: Appendix A (Appendix A), Part C—Declared Problems That Limit PA

The possible limitations were analyzed for 99 patients (47.83% of the total number of returned documents) who indicated at least one limitation from this part of AS. The authors found that the most frequent problems indicated by elderly patients with DM were diseases that limit their PA, fear of exercising alone, lack of motivation, no time because of duties, and, equally, not knowing what exercises they can do and tiredness from daily responsibilities (even if they had time for exercise).

### 3.4. Data Analysis: Relationship between Declared Limitation and Basic Descriptives (Chi-Squared Test)

According to the indicated limitations and data from basic descriptives, females more often indicated that they did not like to exercise alone and had no one who could accompany them (*p* = 0.011). The same problem was indicated by people who were not in a relationship or lived alone in their homes (*p* = 0.05 and *p* = 0.0033, respectively). People with primary education more often stressed that their conditions prevented them from undertaking PA (*p* = 0.046), but people with higher education marked more often that they did not like exercise (*p* = 0.046). Disabled individuals more often indicated disease (*p* = 0.008) and lack of financial resources, necessary in their opinion, for PA (*p* = 0.005). Disease as a limitation also dominated as the choice among people who permanently used any medical supplies (*p* = 0.002).

No further relationship was found between the basic descriptives and proposed barriers that may limit physical activity.

### 3.5. Data Analysis: Appendix A (Appendix A), Part C—Patient’s Opinion on Which Disease Can Limit PA

More than 55% of participants chose at least one disease (not necessarily related to their condition) from the proposed list as a barrier for PA. Joint diseases, as well as diabetes, dominated among these as more than 1/3 of individuals indicated these diseases as an obstacle to taking up exercise. Among all of the ten proposed (disease, symptoms, signs) to be considered as problematic to PA by the responders, the next most commonly selected choices included, in order: leg pain, dizziness, urinary incontinence, excessive body weight, dyspnea, heart diseases, edema, and respiratory diseases.

Only one patient indicated a leg wound as an obstacle for PA.

The analysis showed that people who choose heart disease as a limitation to exercising more often suffered from atherosclerotic heart disease (with or without a history of MI, *p* = 0.005, and *p* = 0.0003, respectively; chi-squared test).

### 3.6. Data Analysis: Result of IPAQ

One hundred fifty-six individuals answered the questions from the IPAQ. The median PA level based on the IPAQ was 1837 MET-min/week (range: 0–9.891). The percentage of people who obtained insufficient (n = 28), sufficient (n = 75), and high levels of PA (n = 53) were 17.95%, 48.08%, and 33.97%, respectively.

The ROC analysis allowed us to determine the BMI value of 33, 3 and age of 75 years as the differentiating point, considering IPAQ scoring (divided into two parts: insufficient vs. at least sufficient PA). People with BMI > 33.3 and >75 years of age (Figure 1 and Figure 2, respectively) more often obtained the value 0—“insufficient level of PA”—on the IPAQ scale; the IPAQ value of 1 or 2 more often related to people ≤ 75 years of age (*p* = 0.044; chi-squared test). There was no difference between men and women with regard to the IPAQ scores.

The rationale selected as a barrier to physical activity did not depend on the IPAQ level.

### 3.7. Data Related to an Insufficient Level of PA (IPAQ = 0)

PCA grouped data from basic descriptives (Appendix A—Appendix A, Part A) related to the value of IPAQ = 0 (insufficient level of physical activity) (Figure 3). The PCA ordered the variables according to how close they were together and thus revealed their relationship. Variables located close to: IPAQ = 0, therefore, showed the patient characteristics most typical of insufficient PA (Figure 3, red circle).

The meta-analysis did not reveal diseases (Appendix A—Appendix A, Part A) more or less related to achieving a score of 0 in the IPAQ (Figure 4). Statistical significance was only demonstrated between “at least a sufficient level of PA” (IPAQ = 1 or 2) and diseases of the digestive system. The same situation was observed for the patients who believed that diabetes could affect taking up PA (Appendix A—Appendix A, Part C)—they more commonly reached “at least sufficient PA” level based on the IPAQ (Figure 4).

## 4. Discussion

Research indicates that only a small percentage, ranging from 2.5 to 22%, of community-dwelling older adults achieve the current World Health Organization (WHO)-recommended PA levels [19,20]. Globally, there are notable age and sex differences in levels of physical inactivity, but after 60 years of age, physical inactivity levels increase in both men and women [16].

We chose to survey individuals over 64 years because the age of 65 is the edge of old age, but also guaranteed that both sexes had reached retirement.

Initially, we assessed the level of PA based on the IPAQ. In the elderly, simple but demanding activities seem to play a greater role than targeted PA. There is a problem as to how to measure the real level of PA in this population because there is no specific survey accepted for the Polish population and dedicated to the elderly that takes into account both everyday PA and leisure time PA [21]. Considering the problems mentioned, the IPAQ was indicated as the best option for comprehensive assessment but was originally validated for the population up to 70, so was appropriate only for the part of the studied population less than 70. To make the assessment uniform, we had to use the IPAQ also for the rest of the studied group, but we encouraged participants to ask us regarding unclear questions. We decided on this method based on conclusions from a British study [22] tailored to the elderly, where the authors stressed that the usefulness of the IPAQ in this group could be strengthened by providing additional details, e.g., about the types and examples of activities.

The IPAQ allows the respondents to be classified into one of three categories of activity: insufficient (less than 600), sufficient (600–1500 or 600–3000), or high (more than 1500 or 3000 MET-min/week). Based on the IPAQ, we found the level of PA was lower (1837 MET-min/week) than in the other study, which was dedicated to the younger population (2079 MET-min/week) [23]. The percentage of people who represented a high level of PA in our study was also lower (27.75%) compared to the younger population from our previous study (37.7%). The PA level results among older people (65+) with DM differed from the similar study results published in 2011 [24]. In that study, 44% of elderly with diabetes had insufficient total PA (defined as the sum of moderate and vigorous PA) based on ADA guidelines and 28% when the Department of Health and Human Services (DHHS) guidelines were applied. Apart from the fact that 13 years have passed since this study, during which time awareness among patients with diabetes may have increased, the American study used different, more rigorous criteria. Therefore, the results of these studies cannot be compared directly.

We found the age of 75 to be a critical point in reaching the “not enough PA level” (more likely at the age over 75). We have not found any studies indicating the age of inflection for this issue. This could be the cue to take more radical actions in the context of individualizing the maintenance of safe PA necessary for the individual’s independence. There is no consensus on the age cut-off for elderly persons, and most researchers accept the threshold of 65 years, but those over 75 years of age are referred to as “late elderly”, which coincides with the cut-off age obtained in our study on insufficient activity level [25].

We also found that among the elderly with DM, a higher BMI is associated with a more frequent choice of any activity at home with the cut-off point of BMI = 28, which was also not explored before. The two-way relationship between body weight and activity should be taken into account here because a higher body weight may be both a consequence of a sedentary lifestyle and a reason for avoiding outdoor activities.

When looking for a potential person among our elderly patients who had an insufficient level of PA, the PCA analysis indicated that it was most often a single woman aged > 75, BMI > 33, living alone, who assessed her level of activity as low in the self-assessment. By identifying such people in the elderly patients’ group, we could help the neediest individuals.

The most important part of the study was to assess the main problems the elderly with DM found to be barriers to being physically active, and the possible relationship between these limitations and basic characteristics. The patients, who had well-balanced glycemia, were overweight rather than obese, with no critical, low value of BMI (<18.5), which could suggest frailty syndrome (FS) based on Women’s Health and Aging Studies (WHAS) [26].

Cardiovascular diseases, as well as musculoskeletal disorders, dominated among the studied group, which is typical for the elderly [27,28]. At the same time, among the obstacles preventing the undertaking of PA, respondents most often chose “diseases”. In the analysis of the last systematic review [16], which was published while our study was still being carried out, we found that medical problems were also pointed out by the elderly in 14 studies that the review covered. It is worth mentioning that none of the studies from the review was conducted in Poland or any Eastern European country. This choice (disease) was particularly popular among disabled people and those using medical supplies, which was also pointed out in some [29,30] but not all [31,32] studies. Our analysis did not confirm these individuals as more connected with an “insufficient” level of PA (based on IPAQ) compared to individuals with no disability. These seemingly contradictory results suggest that although the disabling disease is perceived as a barrier to activity, these individuals made efforts to maintain an appropriate level of daily PA.

The second problem indicated by the surveyed group was the reluctance to undertake exercises alone. This answer choice was most characteristic for women, singles, and those living alone in their homes. The finding about women’s choices aligns with another study (the impact of some cultures or local customs on behavior) [33]. Lack of company, apart from medical problems, was also one of the most important limitations in other studies where the general elderly population was examined [30,34], and walking with a companion was the most preferred among the social exercise preferences [35]. Being lonely can be an obstacle to undertaking PA due to fear of potential events (e.g., falling) with the lack of someone who can help you, as well as due to lack of motivation, which was the third problem indicated by surveyed patients. Many studies have demonstrated the role of falls or fear of falling as a factor limiting undertaking PA in old age [36,37,38]. We did not distinguish this factor from others in the study because fear of falling mainly concerns activities undertaken outside the home [39], while a proper level of daily movement (known as unstructured exercises) can replace at least part of the outside effort [40] by the elderly and is also scored by IPAQ. Support in various activities is vital to human life at every stage. Losing a loved one and not sharing daily duties with family members are significant problems in an aging society. Involving older people in simple household chores can be a source of physical effort that is important for their health and maintaining their independence.

The next most often indicated problem among the studied group was “no motivation”, but also included were lack of time, lack of ideas for PA, and fatigue with everyday responsibilities. Similar to other studies on the general elderly population [30,32,34,36,41,42,43,44], lack of interest in PA was a frequently cited problem. In some studies, a lack of knowledge about exercise (its potential benefits) was explored as a barrier but with no significance on human decisions [43,45], which is opposite to our study results. Perhaps the form of the question regarding knowledge about PA affected the results. We asked specifically if the patient would like to be active but had no idea what exercises they could safely do. Knowledge of the benefits alone is not enough to take action, especially in older people with many objective limitations (such as diseases). Unfortunately, in clinical practice, only this initiative (just information given to the patient) is commonly undertaken [46]. Clear and detailed information, and an established exercise program, ideally supervised, at least initially for some, can motivate older adults [47,48]. Despite the lack of need to work professionally, a lack of time and commitment to daily duties were also selected as obstacles to being physically active. The problems of being “too tired” or “fatigue” were indicated by elderly people even without diabetes previously [32,44]. We have not studied these problems in detail. Still, it can be expected that it may involve taking care of grandchildren or other family members (parents, siblings) for less ill people, as well as frequent medical consultations, preceded by the need to undergo lab tests for the sicker elderly. These duties should be discussed individually, perhaps even with family members, because older people typically may be considered to have a lot of free time and few responsibilities, and should not be tired of everyday life, which is not in line with their view.

In summary, for one in 4–5 patients, at least one of the above four problems (motivation, idea, time, tiredness) was a significant barrier to undertaking more PA. Not all of these problems can be eliminated. Proposing group exercises and making homogenous groups (functionally, based on applicated treatment, e.g., insulin), providing constructive information about possible tailored activities that require sufficient effort could help eliminate problems related to lack of motivation or ideas for exercises, as well as adherence [46]. Regarding this aspect, depression should also be considered as a possible factor [49].

People with higher education indicated that they did not like to be active, which may be due to their poorer condition because they did less physical work in their professional lives. Lack of suitable environmental conditions (e.g., small areas of flat surfaces) was more often indicated by people with only primary education. Although we did not ask about the income of our respondents, it seems reasonable to link primary education to the availability of financial resources. This distinguishes our study population from others. Previous studies from Norway [36], Australia [29], and the United Kingdom [50] did not find cost to be a limitation. Poland differs from the countries mentioned above in terms of the wealth of its inhabitants.

The patient opinions revealed that more than half of the participants indicated at least one of the diseases, signs, or symptoms from the proposed list as a barrier to PA. It is worrying that, although all patients suffered from diabetes, this disease was indicated by one third of them as an obstacle to PA. Taking insulin was not related to the IPAQ’s level of PA. Interestingly, only one person saw a potential problem in activity if there was a leg ulcer. In accordance with the Polish Diabetes Association [51] recommendations, all patients were educated about the role PA plays in the treatment of diabetes and that foot ulceration can be a dangerous complication of the disease. The patients also indicated joint diseases, and almost 40% of patients suffered from them. A study involving a cohort of North American older adults with or at high risk of knee osteoarthritis revealed that only one in five individuals with DM adhered to the recommended guideline of engaging in 150 min of moderate-to-vigorous PA per week, as suggested by the American Diabetes Association [52]. It is a painful disease, so it is not surprising that patients consider it burdensome in the context of making an effort. The same observation applies to heart disease. People who chose this disease as a limitation to exercise more often suffered from atherosclerotic heart disease.

In this field, it is important to point out the limitations of our study. We only examined a population from an urban area (about 87%) and from one diabetic center. However, this could be a valuable source of information about the real problems typical of the large urban elderly population with DM. It is also important to consider the possible variability of the issues reported by patients that may influence their engagement in physical and daily activity.

## 5. Summary

The highlighted problems that impact PA did not differ from those previously indicated in the elderly population without diabetes. According to patient opinion, health status was the main factor limiting physical activity. Other barriers, such as lack of an accompanying person, lack of motivation, or no idea what exercise to undertake, can be associated with loneliness, a common problem described in the literature among the elderly. This distinguishes the elderly population with diabetes from professionally active people if we look at the literature. People over 75 years of age begin to avoid PA more than younger elderly, which leads to an insufficient level of PA. Being more obese means preferring activities at home. In Poland, compared to other European countries, cost plays an important barrier for PA. The study ultimately found that the person with diabetes with the lowest level of PA was likely to be a single, obese woman >75 who was living alone.

The practical dimension of this study emphasizes the role of comorbid conditions, which necessitate the individualization of exercise programs and reinforcing the patient’s belief that such individualization will ensure their safety. Since most physical activities in older age do not require special financial investment, it is important to raise awareness among patients about the benefits of simple activities such as walking or basic exercises that can be performed without specialized equipment. It is worth considering the creation of government-funded support for basic needs related to physical activity, such as comfortable sports shoes, Nordic walking poles, and the organization of activities within housing communities or other local organizations where seniors can exercise in groups. The proposed profile of a person with low physical activity should draw specialists’ attention to individuals with these characteristics, as it is likely that such individuals need special attention and re-education about the role of basic physical activity as a source of health.

## Figures and Tables

**Figure 1 jcm-13-06329-f001:**
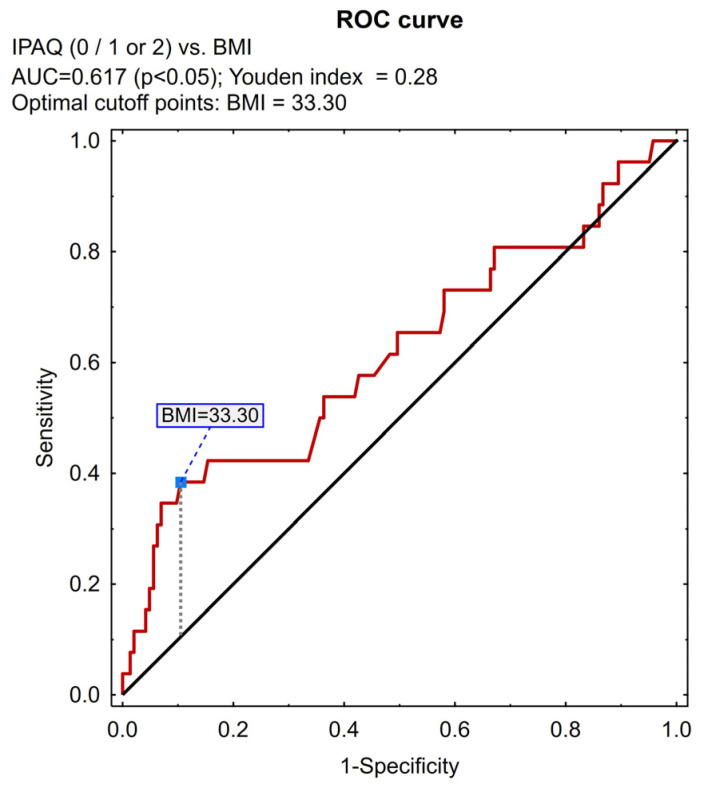
Receiver operating characteristic (ROC) analysis determined for BMI. AUC—area under the curve, BMI—body mass index, IPAQ—international physical activity questionnaire.

**Figure 2 jcm-13-06329-f002:**
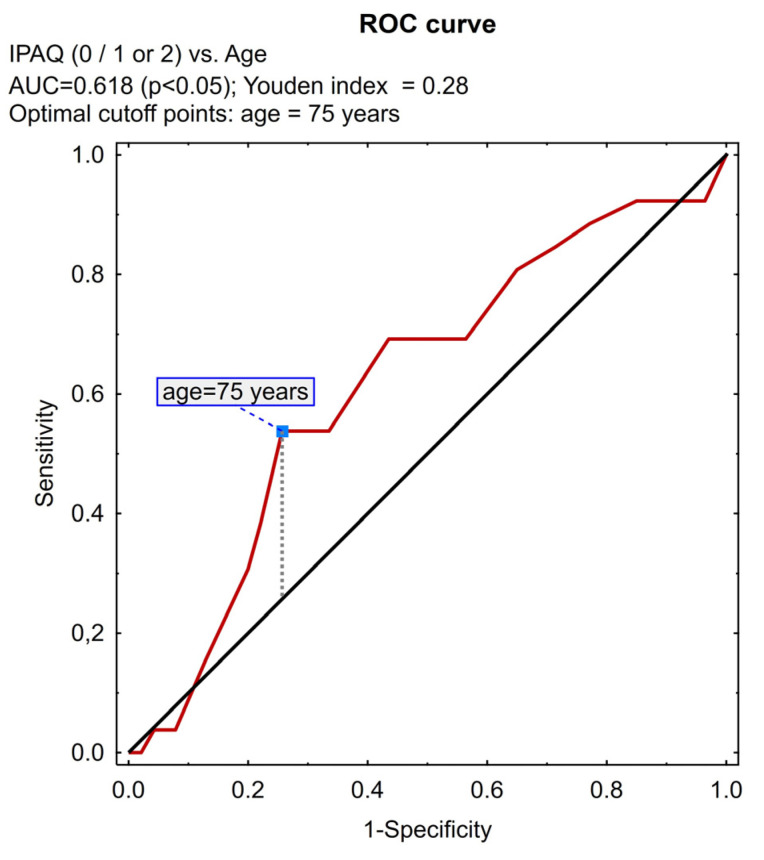
Receiver operating characteristic (ROC) analysis determined for age. AUC—area under the curve, IPAQ—international physical activity questionnaire.

**Figure 3 jcm-13-06329-f003:**
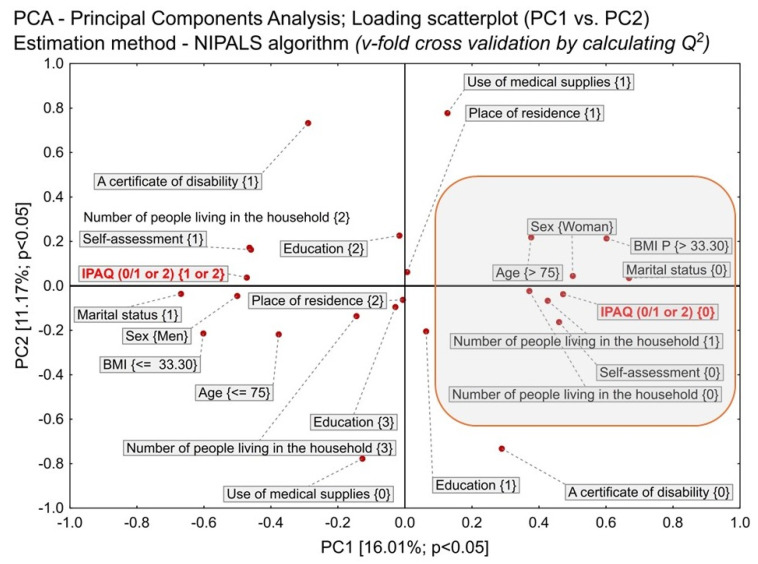
Basic descriptives related to the value of IPAQ = 0 (insufficient level of physical activity). A certificate of disability—{0}: NO, {1}: YES; BMI—body mass index, Education = {1} or {2} or {3}: primary or secondary or higher, respectively; IPAQ—international physical activity questionnaire {0}—insufficient physical activity, {1 or 2}—at least sufficient physical activity; Marital status—{1} or {0}: in a relationship or single, respectively; Number of people living in the household—{1} or {2} or {3} or {0}: means “one” or “two” or “three or more” people or “no one”, respectively; Place of residence—{1} other than big city or {2}: big city; Self-assessment—{0}-insufficient physical activity in patient’s opinion, {1}—at least sufficient physical activity in patient’s opinion; “Use of medical supplies”—{0}: NO, {1}:YES.

**Figure 4 jcm-13-06329-f004:**
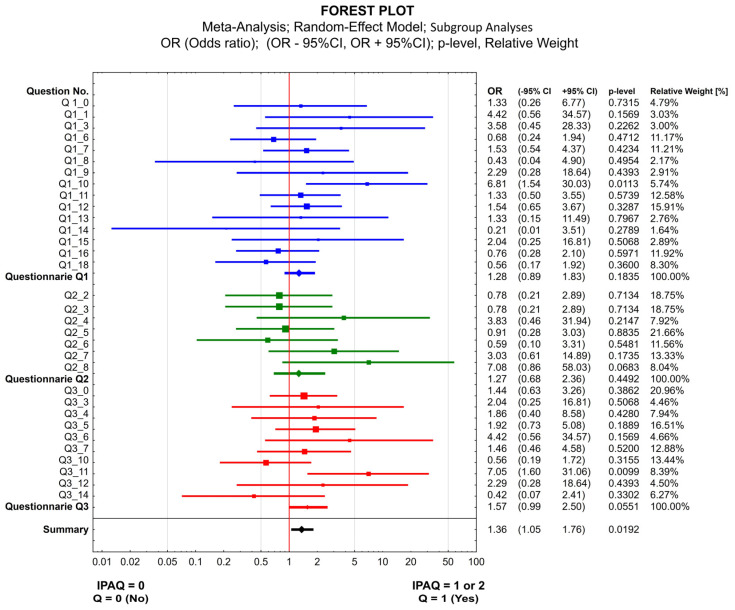
Correlations between the IPAQ variable (0 vs. 1 or 2) and the results of the accompanying survey. IPAQ—international physical activity questionnaire (0-infufficient, 1 or 2—at least sufficient level of physical activity); Questionnaire 1 (epidemiological): Q 1_0: Do you chronically suffer from the following health problems? Q1_1: Ischemic Heart Disease and Myocardial Infarction; Q1_3: Atrial Fibrillation; Q1_6: Hypertension; Q1_7: Chronic Venous Disease; Q1_8: Embolism or Thrombosis requiring long-term treatment; Q1_9: Lung or Bronchial Disease; Q1_10: Gastrointestinal Disease; Q1_11: Diseases of the Kidneys, Prostate gland, Urinary bladder; Q1_12: Diseases of the Musculoskeletal System; Q1_13: Cancer under treatment; Q1_14: Female Reproductive System Diseases; Q1_15: Skin Diseases; Q1_16: Thyroid Gland Diseases; Q1_18: Other chronic diseases; Questionnaire 2 (limitations in engaging in physical activity): Q2_2 I don’t have time—for other reason than work; Q2_3 I feel tired by the duties I have to do every day, even though I have time [C]; Q2_4 I have time but have no idea what I could do; Q2_5 I have diseases that limit my physical activity; Q2_6 I don’t like physical activity, but I would like to undertake it; Q2_7 I can’t find the motivation; Q2_8 I don’t like to exercise alone; Questionnaire Q3 (patient’s opinion about the limitations): Q3_0—Do you think that the following disease (or symptoms) may be an obstacle to physical activity?; Q3_3—Oedema; Q3_4—Obesity; Overweight; Q3_5—Joint disease; Q3_6—Dizziness; Q3_7—Leg pain; Q3_10—Urinary Incontinence; Q3_11—Diabetes; Q3_12—Varicose veins; Q3_14—Other diseases. Note: In the presented forest plot, questions in which the two-way contingency table (2 × 2 table) based on the response distribution contained a value of 0 in any of the 4 cells were not included, as it is not possible to calculate the odds ratio (OR) in such situations.

## Data Availability

The data presented in this study are available on request from the corresponding author (edyta.sutkowska@umw.edu.pl).

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
