# Peer review of "The Level and Limitations of Physical Activity in Elderly Patients with Diabetes"

_jcm, 2024, doi:10.3390/jcm13216329_

Round 1
Reviewer 1 Report
Comments and Suggestions for Authors
Dear Authors,
Thank you for your manuscript. Please see my comments below.
Please move the study aim from Methods to the end of the Introduction. If you would like to highlight this part, it is recommended to write a separate section at the end of the Introduction "The present study".
Sorry, if I missed that but sample characteristics and the recruitment procedures of the study subjects are missing. The methods section is recommended to be organized according to STROBE guidelines as follows: 1. Study design. 2. Participants. 3. Settings and Procedure. 4. Study Measures. 5. Sample size. 6. Statistical Analysis. The study measures (e.g., questionnaires or single questions used) should be described in greater detail.
However, my major concern is regarding statistical analysis and the complicated presentation of the data. Figure's 1 quality is poor and it is not possible to read the text inside it. In Figure 2 (meta-analysis), the ORs are reported without sufficient explanation of what they mean in the context of this study. Also, the presentation is unclear as the meaning of Q1_0, Q1_1, etc., are not explained to the reader. The Figure should be independent of the results description or supplementary files and should be self-explanatory.
The paper mentions that a meta-analysis model was used to analyze relationships between the IPAQ results and the accompanying survey results based on odds ratios (ORs) with 95% confidence intervals (CIs). However, this does not fit the typical definition of a meta-analysis. Meta-analysis traditionally refers to a statistical technique that combines results from multiple independent studies to synthesize evidence on a particular topic. Here, the authors are using a single dataset and analyzing relationships between variables within that dataset. This is not a meta-analysis but rather an analysis of associations within their study population. Using binary logistic regression would be more appropriate. This approach would allow for modelling the likelihood of insufficient physical activity based on age and BMI, potentially with these variables categorized into smaller intervals. This would give a clearer understanding of the risk associated with different levels of age and BMI and the significance of these factors in predicting physical activity.
Next, the ROC analysis was applied to determine cutoff values for age (75 years) and BMI (33 kg/m²) to predict insufficient physical activity. The ROC analysis is typically used to evaluate the performance of diagnostic tests, comparing true positive rates (sensitivity) and false positive rates (specificity) to identify optimal thresholds for decision-making. It is often used when distinguishing between two groups (e.g., disease vs. no disease/controls) based on a test or scale. In this case, while the use of ROC analysis might show some discriminative ability of age and BMI for predicting physical activity or inactivity, a more suitable statistical method would indeed be a logistic regression analysis, again. Importantly, the cut-off values determined using ROC analysis for BMI and age (BMI >33 kg/m² and age >75 years) are specific to this study sample and may not be applicable to other populations or settings. Logistic regression could model the probability of insufficient physical activity as a function of age and BMI and could include interaction terms or categorize these variables into meaningful groups to see their predictive effects.
Moreover, the use of PCA is mentioned to reveal relationships between key variables. However, PCA is typically used for dimensionality reduction and identifying patterns in data with many correlated variables. The description of how PCA was used to group characteristics associated with insufficient physical activity is somewhat unclear and may not be the best approach for this type of analysis. A more detailed explanation of the PCA methodology and its justification is needed.
Lastly, the paper does not seem to account for potential confounding variables comprehensively. Factors such as socio-economic status, duration of diabetes, presence of depression, and medication use could influence physical activity levels and should be controlled for in the analysis to avoid biased results.
To summarize, I recommend:
- Re-analyze the data using logistic regression or another suitable method to identify significant predictors of physical activity levels.
- Consider including a multivariate analysis that controls for confounders such as socio-economic status, diabetes duration, and comorbidities.
- Provide a more detailed justification for their use of statistical methods, such as PCA.
- Improve the clarity and detail of figures.
- Use correct statistical and methodological terminology and avoid causal language when discussing associations found in a cross-sectional study.
- Discuss limitations more thoroughly, including biases from self-reported data, generalizability, and potential confounders not accounted for.
Moderate English revision is required after essential corrections on the manuscript.
Author Response
07-10-2024
Dear Editor and Reviewers
We would like to thank both Reviewers and Editor for their detailed reviews and comments. Where possible, we have addressed them below, either by implementing the recommended changes or by explaining our reservations regarding the reviewers' suggestions. We understand that some of the comments may stem from subjective preferences, and differences may arise not only between the author and reviewer but also among the reviewers themselves. It is natural that each of us has our own experiences and ways of conveying information that resonate with us. Therefore, we kindly ask for understanding if we are unable to fully incorporate all of the reviewers' suggestions. However, we have made an effort to explain why, in our opinion, some of the suggestions were not included.
All changes in the text have been marked in red.
Reviewer no 1
Dear Authors,
Thank you for your manuscript. Please see my comments below.
- Please move the study aim from Methods to the end of the Introduction. If you would like to highlight this part, it is recommended to write a separate section at the end of the Introduction "The present study".
The information about the aim of the study has been moved to the Introduction section.
- Sorry, if I missed that but sample characteristics and the recruitment procedures of the study subjects are missing.
The 3.1 section is dedicated to the sample characteristics, and as mentioned in this section, some of them are included in the table provided in Supplement no. 2 ( part A of the table) due to the large volume of data .
- The methods section is recommended to be organized according to STROBE guidelines as follows: 1. Study design. 2. Participants. 3. Settings and Procedure. 4. Study Measures. 5. Sample size. 6. Statistical Analysis.
The article includes all the necessary study elements required by the STROBE guidelines: study design, participants, and settings are covered in section 2.2, while procedure and study measures are described in section 2.1. The statistical methods used are explained in a separate section. Please note that this sequence was chosen to maintain a smooth flow of the narrative. For example, it would be difficult to discuss surveys in section 2.1 without first explaining their purpose. Therefore, procedure and study measures were clarified first, making it easier to understand the subsequent methodological parts.
- The study measures (e.g., questionnaires or single questions used) should be described in greater detail.
The article contains a wealth of information, and the analysis and discussion took up considerably more space than typically allowed by the editorial guidelines. We are grateful that the article can remain in its original, longer form. Explaining the IPAQ questionnaire and the AS questionnaire in detail would turn the article into a book chapter and, in our opinion, make it difficult for the reader to follow. The questions from the AS questionnaire are included in Supplement no. 1 and are presented in the same way patients used them. For the IPAQ questionnaire, the source literature is provided so that those unfamiliar with it can refer to that material. If the editorial team agrees and you find it necessary to explain the IPAQ questionnaire, we will of course describe its contents. However, please keep in mind our concern regarding the length of the text.
- However, my major concern is regarding statistical analysis and the complicated presentation of the data.
5.1. Figure's 1 quality is poor and it is not possible to read the text inside it.
We hope that the improved quality is now more readable.
5.2. In Figure 2 (meta-analysis), the ORs are reported without sufficient explanation of what they mean in the context of this study. Also, the presentation is unclear as the meaning of Q1_0, Q1_1, etc., are not explained to the reader. The Figure should be independent of the results description or supplementary files and should be self-explanatory.
We sincerely apologize for this oversight – the person submitting the article missed the document containing the index.
5.3. The paper mentions that a meta-analysis model was used to analyze relationships between the IPAQ results and the accompanying survey results based on odds ratios (ORs) with 95% confidence intervals (CIs). However, this does not fit the typical definition of a meta-analysis. Meta-analysis traditionally refers to a statistical technique that combines results from multiple independent studies to synthesize evidence on a particular topic. Here, the authors are using a single dataset and analyzing relationships between variables within that dataset. This is not a meta-analysis but rather an analysis of associations within their study population. Using binary logistic regression would be more appropriate. This approach would allow for modelling the likelihood of insufficient physical activity based on age and BMI, potentially with these variables categorized into smaller intervals. This would give a clearer understanding of the risk associated with different levels of age and BMI and the significance of these factors in predicting physical activity.
All variables included in the meta-analysis (both the dependent variable – IPAQ=0 vs. IPAQ=1 or 2, and the independent variables, which were the answers to individual questions from questionnaires Q1, Q2, and Q3) were dichotomous. In such cases, the results obtained using the meta-analysis model are identical to those obtained after conducting univariate logistic regression, as they directly result from the significance assessment of the chi-squared (χ²) statistic calculated for the constructed 2x2 contingency tables. The choice of the meta-analysis model was motivated by the possibility of visualizing the results in a forest plot, which, in the authors' opinion, provides a clearer form of presentation compared to the traditional tabular format. The random-effects meta-analysis model used, which does not assume homogeneity of the distributions of the analyzed variables, is general enough that it is often applied to the analysis of data other than randomized clinical trials in evidence-based medicine (EBM).
The authors also analyzed the data using multivariate logistic regression during the analysis process; however, the results obtained were consistent with those presented in the paper. They were not included as they did not add any additional value to the conclusions already drawn.
5.4. Next, the ROC analysis was applied to determine cutoff values for age (75 years) and BMI (33 kg/m²) to predict insufficient physical activity. The ROC analysis is typically used to evaluate the performance of diagnostic tests, comparing true positive rates (sensitivity) and false positive rates (specificity) to identify optimal thresholds for decision-making. It is often used when distinguishing between two groups (e.g., disease vs. no disease/controls) based on a test or scale. In this case, while the use of ROC analysis might show some discriminative ability of age and BMI for predicting physical activity or inactivity, a more suitable statistical method would indeed be a logistic regression analysis, again. Importantly, the cut-off values determined using ROC analysis for BMI and age (BMI >33 kg/m² and age >75 years) are specific to this study sample and may not be applicable to other populations or settings. Logistic regression could model the probability of insufficient physical activity as a function of age and BMI and could include interaction terms or categorize these variables into meaningful groups to see their predictive effects.
The authors are fully aware that ROC analysis was historically developed to evaluate the quality of diagnostic tests. However, its mathematical structure makes it the most optimal technique for determining a cutoff point for a continuous variable in such a way that it divides the variable into two maximally differentiated subgroups, defined by an additional dichotomous variable. This procedure is so general and efficient that it is now commonly included in statistical software that allows for logistic regression analysis.
The issue of evaluating how much the cutoff points determined using ROC analysis in the study refer to the sample versus the general population seems to depend on factors other than the choice of a particular statistical technique. In the authors' opinion, ensuring the homogeneity and representativeness of the study sample is achieved at the stage of random selection from the general population, and the choice of statistical techniques used in the data analysis plays a marginal role in this regard.
5.5. Moreover, the use of PCA is mentioned to reveal relationships between key variables. However, PCA is typically used for dimensionality reduction and identifying patterns in data with many correlated variables. The description of how PCA was used to group characteristics associated with insufficient physical activity is somewhat unclear and may not be the best approach for this type of analysis. A more detailed explanation of the PCA methodology and its justification is needed.
The use of Principal Component Analysis (PCA) in this study aimed to extract and visualize a set of characteristics associated with individuals with low physical activity on a single graph. The use of PCA in the study was primarily illustrative, as this technique is often classified as a method based on dimensionality reduction. It allows for the representation of mutual correlations between the analyzed variables on a 2D graph and assesses their strength, understood as the percentage contribution of each individual variable to explaining the total variability contained in the experimental matrix.
To assess the statistical significance of the correlations identified through PCA among the individual variables, other statistical tests were employed. Dimensionality reduction techniques, while not providing answers to the question of which correlations are statistically significant, have no alternative in answering where to look for statistically significant relationships. This is precisely the role they played in this study. It should be noted that although Principal Component Analysis (PCA) is based on the same computational technique as standard Multidimensional Scaling (MDS) techniques, namely the decomposition of the experiment matrix according to singular values (SVD, Singular Value Decomposition), classifying it within this group of analyses may be problematic. The basis for calculations in MDS methods is the experiment matrix defined as a distance matrix between the analyzed variables. The distance between two variables is defined by the metric used, which has a very precise axiomatic definition. The most commonly used metrics include Euclidean, Minkowski, Manhattan, and Chebyshev metrics. In PCA, the experiment matrix is a correlation matrix between variables, and correlation coefficients do not meet the definition of a metric and cannot be treated as distances.
5.6. Lastly, the paper does not seem to account for potential confounding variables comprehensively. Factors such as socio-economic status, duration of diabetes, presence of depression, and medication use could influence physical activity levels and should be controlled for in the analysis to avoid biased results.
We did not assess the direct impact of diabetes duration on physical activity (PA). However, the duration of diabetes can affect patients' engagement in and level of physical activity, including daily activities. This occurs for obvious reasons: as diabetes progresses, complications like neuropathy, cardiovascular issues, or musculoskeletal problems may arise, making physical activity more challenging. Additionally, long-term diabetes can lead to reduced motivation or physical ability to engage in regular exercise. Nonetheless, our assessment covered the variables mentioned above that are influenced by the duration of the disease, such as comorbidities that develop with age and decreased motivation. The duration of the disease per se does not translate to activity.
The socio-economic status, to what extent its assessment was enabled by the information from the survey (intermediate assessment), is discussed in lines: 474-477.
The involvement of depression is mentioned in line 469 (literature no. 55).
The results are presented only for the relationships for which statistical significance was demonstrated. For obvious reasons, the medication mentioned, regardless of the obtained relationship (no such relationship was demonstrated), was insulin (the fear of hypoglycemia when using it may discourage physical activity) – see line 485.
5.7. To summarize, I recommend:
- Re-analyze the data using logistic regression or another suitable method to identify significant predictors of physical activity levels.
- Consider including a multivariate analysis that controls for confounders such as socio-economic status, diabetes duration, and comorbidities.
- Provide a more detailed justification for their use of statistical methods, such as PCA.
- Improve the clarity and detail of figures.
The explanatory information is provided above.
- Use correct statistical and methodological terminology and avoid causal language when discussing associations found in a cross-sectional study.
I'm sorry, but I don't understand this recommendation. As mentioned at the beginning, the translation is the responsibility of a native speaker with medical experience. We only discuss correlations where they were assessed; if we don't use the term "correlation" and only use "relationship," it means that correlations were not examined, but rather dependencies, e.g., in PCA.
- Discuss limitations more thoroughly, including biases from self-reported data, generalizability, and potential confounders not accounted for.
Please note that the limitations are not only at the end of the discussion (line 501) but also in lines 333-348 (regarding IPAQ) and 377-381 (regarding BMI), directly after discussing these variables or results. The sequence has been added to the main text: “It is also important to consider the possible variability of the issues reported by patients that may influence their engagement in physical and daily activity.”
5.8. Moderate English revision is required after essential corrections on the manuscript.
All texts we prepare must (a requirement imposed by the university) be edited. This article has also been proofread by a native speaker from a company specializing in medical text editing, and the certificate has been attached for the editor's reference. We kindly request that any noticed errors be pointed out, as this is the only way we can promote the aforementioned service.
With the hope that we have clarified your doubts.
Team Leader- on behalf of the study Team and university statistics Team.
ES
Reviewer 2 Report
Comments and Suggestions for Authors
The manuscript by Biernat et al., titled: "The level and limitations of physical activity in elderly patients with diabetes" is an interesting study investigating the the extent to which physical activity is impacted by the condition of diabetes in elderly patients. This is an interesting topic with potential clinical/public health implications.
The reviewer would like to raise the following points for the authors' consideration:
1. BMI does not have units. The kg/m2 is a calculation not a unit of a measure.
2. Please identify and report explicitly the inclusion and exclusion criteria for the study conducted.
3. What was the rationale for the determination of the specific number of participants? (power calculation).
4. How were potential confounding factors identified and treated? For example smoking status, different medication regimes, different effectiveness on diabetes management and most importantly progression and time with the disease.
5. Glycosylated hemoglobin must be reported in %. It represents an overtime condition. It is meaningless to convert this into concentration of glucose because the measurement represents a condition over time that is actually used to assess management. There is no specific concentration of glucose that is associated physiologically with a given % for glycosylated hemoglobin (HbA1c).
6. Ware the survey tools used in the study (questionnaires) validated in their used form for the population they were used on?
7. The graphs/figures are blurry and apparently not in high resolution thus making it difficult to read and interpret by the reviewer (especially Figure 1).
8. Consider a different structure with the graphs inserted at more appropriate places within the text.
9. Consider including a strengths and limitations section clearly delineated at the end of the discussion section.
10. In the findings and in the discussion authors elude to the fact that the disease and the lack of companionship are the main reasons preventing the patients from PA. How did they separate between the two? Could there be a compounding effect?
11. Did the authors consider body composition? This is a much stronger predictor for physical activity due to the elements of bone density and muscle mass. As such the BMI is a very crude and most likely unsuitable measure to associate with PA especially at this age group (elderly).
Comments on the Quality of English LanguageEnglish is OK but the manuscript would benefit from a read through by a native English speaker.
Author Response
07-10-2024
Dear Editor and Reviewers
We would like to thank both Reviewers and Editor for their detailed reviews and comments. Where possible, we have addressed them below, either by implementing the recommended changes or by explaining our reservations regarding the reviewers' suggestions. We understand that some of the comments may stem from subjective preferences, and differences may arise not only between the author and reviewer but also among the reviewers themselves. It is natural that each of us has our own experiences and ways of conveying information that resonate with us. Therefore, we kindly ask for understanding if we are unable to fully incorporate all of the reviewers' suggestions. However, we have made an effort to explain why, in our opinion, some of the suggestions were not included.
All changes in the text have been marked in red.
Reviewer no 2.
The manuscript by Biernat et al., titled: "The level and limitations of physical activity in elderly patients with diabetes" is an interesting study investigating the extent to which physical activity is impacted by the condition of diabetes in elderly patients. This is an interesting topic with potential clinical/public health implications.
The reviewer would like to raise the following points for the authors' consideration:
- BMI does not have units. The kg/m2 is a calculation not a unit of a measure.
Thank you for your feedback; you are absolutely correct. We have corrected it by removing "kg/m²." We apologize for any confusion, but in some of our previous work, the reviewers requested that we include information about the calculation.
- Please identify and report explicitly the inclusion and exclusion criteria for the study conducted.
The inclusion and exclusion criteria are in chapter 2.2, lines: 104-114. Please do not hesitate to ask if something is not clear.
- What was the rationale for the determination of the specific number of participants? (power calculation).
The basis for calculating the sample size was the evaluation of the power of the chi-squared statistic for 2x2 tables, with a power of no less than 0.8. The sample was estimated at approximately 190 people.
- How were potential confounding factors identified and treated? For example smoking status, different medication regimes, different effectiveness on diabetes management and most importantly progression and time with the disease.
All necessary data were obtained from the AS survey completed by the patient. Variables such as smoking status or various treatment regimens were not analyzed, except for the information regarding insulin therapy, as its use can significantly impact the level of physical activity (concern about hypoglycemia). For this reason, despite the lack of influence of this therapy on activity and issues with engaging in it in our study, it was mentioned in the discussion of the results.
The assessment of glycemic control was analyzed through HbA1c, and body mass through BMI. Neither blood pressure values nor lipid profile values were analyzed. However, the AS included questions about hypertension (YES/NO) or medications, including those improving lipid profiles. None of these had a relationship with the level of physical activity or reported issues in engaging in it (in the text of the paper, only those values for which relationships were demonstrated were discussed, except for insulin, which was explained above).
I am not sure what you mean by “disease progression”—but if you are referring to complications here is our explanation: Epidemiological questions inquired about diseases of the urogenital system (including kidney disease), cardiovascular diseases, and "other chronic diseases." No relationship was found for any of these (among potential diabetic complications) with the level of activity (please see line 268: “No further relationship was found between the basic descriptives and proposed barriers that may limit physical activity”.). The duration of the disease was not recorded. However, the duration of diabetes can affect patients' engagement in and level of physical activity, including daily activities. This occurs for obvious reasons: as diabetes progresses, complications like neuropathy, cardiovascular issues, or musculoskeletal problems may arise, making physical activity more challenging. Additionally, long-term diabetes can lead to reduced motivation or physical ability to engage in regular exercise. Nonetheless, our assessment covered the variables mentioned above that are influenced by the duration of the disease, such as comorbidities that develop with age and decreased motivation. The duration of the disease per se does not translate to activity.
- Glycosylated hemoglobin must be reported in %. It represents an overtime condition. It is meaningless to convert this into concentration of glucose because the measurement represents a condition over time that is actually used to assess management. There is no specific concentration of glucose that is associated physiologically with a given % for glycosylated hemoglobin (HbA1c).
I apologize, but I do not understand this comment. We provided HbA1c in % and mmol/mol because this is required by most journals. We did not convert HbA1c values to average glucose levels! Since 2010, the World Health Organization (WHO) has recommended reporting HbA1c results in mmol/mol as well as in percentages to ensure consistency and facilitate easy comparison of results in various clinical and research contexts.
- Ware the survey tools used in the study (questionnaires) validated in their used form for the population they were used on?
The tool named AS (Accompanying Survey) is not actually a survey but rather a collection of questions regarding demographic and epidemiological data, as well as questions about daily problems that hinder physical activity. It does not have a scoring scale and does not require validation. Nevertheless, to assess and enhance the comprehensibility of the questions, two "resolutions" were used. When used for the first time patients were asked to complete it again within 7 days. The aim was to test the reliability of this tool to minimise the risk of randomisation of responses (literature no 23). Additionally as was mentioned in Metodology section: One month of community consultation at the Diabetic Center preceded the selection of the proposed list of limitations to create this last part of the AS.
- The graphs/figures are blurry and apparently not in high resolution thus making it difficult to read and interpret by the reviewer (especially Figure 1).
We hope that the improved quality is now more readable
- Consider a different structure with the graphs inserted at more appropriate places within the text.
I'm sorry, but we do not have any influence over where the figures that are supplements are placed. The other two are located according to the editor's instructions in the places where they are referenced in the text.
- Consider including a strengths and limitations section clearly delineated at the end of the discussion section.
Please note that the limitations (and strengths) are not only at the end of the discussion (line 501) but also in lines 333-348 (regarding IPAQ) and 377-381 (regarding BMI), directly after discussing these variables or results. Due to the current length of the text, we believe that summarizing them again at the end would unnecessarily extend the article. Of course, if you think this is necessary and the editor agrees to such an expansion of the text, we will include their "summary" at the end.
The sequence has been added to the main text: “It is also important to consider the possible variability of the issues reported by patients that may influence their engagement in physical and daily activity.”
- In the findings and in the discussion authors elude to the fact that the disease and the lack of companionship are the main reasons preventing the patients from PA. How did they separate between the two? Could there be a compounding effect?
This was assessed through a meta-analysis presented in Figure 2. Once again, I apologize for the lack of explanations under this figure regarding Q. Including now these explanations will surely facilitate understanding. PCA was used to cluster demographic and epidemiological variables together. Clustering features refers to "our patient," meaning a set of variables that most frequently coexist in a person with low physical activity : ” The study ultimately found that the person with diabetes with the lowest level of PA was likely to be a single, obese woman >75 who was living alone”.
- Did the authors consider body composition? This is a much stronger predictor for physical activity due to the elements of bone density and muscle mass. As such the BMI is a very crude and most likely unsuitable measure to associate with PA especially at this age group (elderly).
That is, of course, a valid point; however, we did not include body composition because it is not a component of everyday medical practice. We aimed to identify those characteristics that are typically assessed during a patients’ visit.
- English is OK but the manuscript would benefit from a read through by a native English speaker.
All texts we prepare must (a requirement imposed by the university) be edited. This article has also been proofread by a native speaker from a company specializing in medical text editing, and the certificate has been attached for the editor's reference. We kindly request that any noticed errors be pointed out, as this is the only way we can promote the aforementioned service.
With the hope that we have clarified your doubts.
On behalf of the team
Team Leader- ES
Round 2
Reviewer 1 Report
Comments and Suggestions for Authors
Dear Authors,
Thank you for the corrections made.
Please provide all the questions used in the study questionnaire in the Methods section, along with the response options and references supporting their validity. Only the IPAQ is described now.
Also, at the end of the discussion, please provide practical implications of your study.
Author Response
Rev no 1/ round 2:
We are pleased that our explanations from the previous letter were understood and accepted. In response to the current request: in the manuscript, we have highlighted the added text in green (previous changes from the last review remain in red). Including all sentences from the "Limitations" section of the AS in the manuscript seems impractical due to their length, especially since they are available in two documents attached as Supplements. This document (Supplement) contains both the original AS text provided to patients and a table with the results, which includes the full questions. To facilitate understanding, we have added more detailed information in the Methodology section.
Q1: Please provide all the questions used in the study questionnaire in the Methods section, along with the response options and references supporting their validity. Only the IPAQ is described now.
As mentioned in the Methodology, we created the list of "limitations" based on community consultations. The tool named AS (Accompanying Survey) is not actually a survey but rather a collection of questions regarding demographic and epidemiological data, as well as questions about daily problems that hinder physical activity. It does not have a scoring scale and does not require validation. Nevertheless, to assess and enhance the comprehensibility of the questions, two "resolutions" were used. When used for the first time patients were asked to complete it again within 7 days. The aim was to test the reliability of this tool to minimise the risk of randomisation of responses (literature no 23). Additionally as was mentioned : “One month of community consultation at the Diabetic Center preceded the selection of the proposed list of limitations to create this last part of the AS.”
However, the literature we refer to in the Discussion relates in details to questions about physical activity limitations used in other studies, as well as the barriers confirmed in those studies (references no: 16,29-40, 43-48,51,54). Please note that creating our own list of "limitations" proved to be an appropriate choice, as one of the issues identified in our studied population was “financial constraints”. This distinguished the Polish elderly population from those in other countries that have undertaken similar studies (lack of financial resources, deemed necessary for PA by participants (p=0.005). Therefore, if we had included only the questions used in other studies in our "Limitations list," focusing solely on those for which statistical significance was demonstrated previously, we would have overlooked this important (and practically noticeable) issue. This point is also mentioned in the Discussion section.
Q2: Also, at the end of the discussion, please provide practical implications of your study.
Has been added.
Thank you for your detailed assessment and inspiration.
Kind regards
Team leader: ES
Reviewer 2 Report
Comments and Suggestions for Authors
The authors have reasonably addressed the reviewer's points. Proofreading is suggested.
Author Response
We would like to thank the Reviewer for his insightful assessment.